# On the Emergence of Linear Analogies in Word Embeddings

**Daniel J. Korchinski**[*]
Department of Physics
Ecole Polytechnique Fédérale de Lausanne (EPFL)
Lausanne, VD Switzerland
daniel.korchinski@epfl.ch

**Dhruva Karkada**
Department of Physics
UC Berkeley
Berkeley, CA, USA
dkarkada@berkeley.edu

**Yasaman Bahri**
Google DeepMind
Mountain View, CA, USA
yasamanbahri@gmail.com

**Matthieu Wyart**
Johns Hopkins & EPFL
Baltimore, MD, USA & Lausanne, VD Switzerland
mwyart1@jh.edu

## Abstract

Models such as Word2Vec and GloVe construct word embeddings based on the co-occurrence probability $P(i, j)$ of words $i$ and $j$ in text corpora. The resulting vectors $W_i$ not only group semantically similar words but also exhibit a striking linear analogy structure—for example, $W_{\text{king}} - W_{\text{man}} + W_{\text{woman}} \approx W_{\text{queen}}$—whose theoretical origin remains unclear. Previous observations indicate that this analogy structure: (i) already emerges in the top eigenvectors of the matrix $M(i, j) = P(i, j)/P(i)P(j)$, (ii) strengthens and then saturates as more eigenvectors of $M(i, j)$, which controls the dimension of the embeddings, are included, (iii) is enhanced when using $\log M(i, j)$ rather than $M(i, j)$, and (iv) persists even when all word pairs involved in a specific analogy relation (e.g., king–queen, man–woman) are removed from the corpus. To explain these phenomena, we introduce a theoretical generative model in which words are defined by binary semantic attributes, and co-occurrence probabilities are derived from attribute-based interactions. This model analytically reproduces the emergence of linear analogy structure and naturally accounts for properties (i)–(iv). It can be viewed as giving fine-grained resolution into the role of each additional embedding dimension. It is robust to various forms of noise and agrees well with co-occurrence statistics measured on Wikipedia and the analogy benchmark introduced by Mikolov et al.

## 1. Introduction and Motivation

Vector representations of words have become a cornerstone of modern natural language processing. Models such as Word2Vec [1, 2] and GloVe [3] map any word $i$ to some continuous vector space $W_i \in \mathbb{R}^K$ based on their co-occurrence statistics in large text corpora. These embeddings capture rich semantic relationships, including a remarkable form of analogical structure: for instance, the arithmetic expression

$$W_{\text{king}} - W_{\text{man}} + W_{\text{woman}} \approx W_{\text{queen}} \tag{1}$$

often holds in embedding space [4, 2]. While widely observed, understanding the origin of this geometric regularity remains a challenge. Several works [5, 3] indicate that this property is already contained in the co-occurrence matrix $M(i, j) = P(i, j)/P(i)P(j)$, or in $PMI(i, j) \equiv \log(M(i, j))$, called

---

[*]https://djkorchinski.github.io/

the pointwise mutual information or PMI matrix. Here, $P(i,j)$ is the empirical probability that words $i$ and $j$ appear together within a context window, and $P(i)$ is the marginal probability of word $i$.

Key observations to explain are that (i) the top eigenvectors of $M$ already separate semantic aspects of words [6]; (ii) If the word embedding is built by approximating $M \approx W^\top W$, where $W$ is a $K \times m$ matrix whose columns $W_i$ are the embeddings of each word $i = 1...m$ and rows $W_i^\top$ are the (scaled) eigenvectors of $M$, then one finds that linear analogies initially becomes more accurate as $K$ increases, such that more of the top eigenvectors of $M$ are retained [6]; (iii) linear analogies improve further when the PMI is used instead of $M$ itself, i.e. by approximating the $PMI \approx W^\top W$ [3, 7]. Moreover, (iv) linear analogies persists even when the word pairs exemplifying the target analogy are removed from the dataset [8] (e.g. removing from the corpus all word pairs including a masculine word and its feminine counterpart, such as man-woman, king-queen, etc.).

## 2. Our Contributions

In this work, we propose a theoretical generative model for word co-occurrences, which rationalizes the observations above. The model enables us to compute the properties of $M$ and of the PMI analytically and elucidates how linear analogies emerge.

(1). Our theory posits that each word is defined by a vector of $d$ discrete semantic attributes, and that each attribute of a word affects its context in an independent manner. This leads to an exact Kronecker product structure in the co-occurrence matrix $P(i,j)$, and derived quantities such as $M(i,j)$, which allows us to compute eigenvectors and eigenvalues of $M$ and the PMI analytically.

(2). We show that analogies naturally emerge as a result of dominant eigenvectors when the matrix $M$ or $\log(M)$ is used to create embeddings: vector arithmetic on the $d$-dimensional space of attributes gives rise to the same arithmetic relations for word embeddings. We emphasize that this property arises naturally in the eigendecomposition as a result of the factorization assumption. We show that this linearity however breaks down as $K$ increases if $M$ is considered instead of the PMI.

(3). We find that our conclusions are remarkably robust to the presence of noise or to perturbations of the co-occurrence matrix, including the addition of i.i.d. noise, the pruning of most words, as well as the removal of co-occurrence probabilities that compose an analogy (such as "king"-"man" and "queen"-"woman"), and correlations between attributes, showing how the analogy-supporting subspace remains robust.

(4). Throughout this work, we test specific predictions of our theoretical model numerically through comparison with Wikipedia text, demonstrating the predictive power of our model.

## 3. Related Work

Extensive empirical evidence suggests that many models of natural language [2, 3], including those trained on non-english corpora [9–11], and contemporary large language models [12–15], exhibit striking linear structure in their latent space. This observation motivates contemporary research in modern language models, including mechanistic interpretability [16–18], in-context-learning [19–21], and LLM alignment [22–24]. Linear analogy structure in word embedding models is the natural precursor to these phenomena; thus, to understand linear representations in general, it is important to develop a theoretical understanding of linear analogies in simple models.

The origin of the linear analogy structure in word embedding models has been the subject of intense study [25–29]. Prior works [3, 27, 30] have based their reasoning on the insight that ratios of conditional probabilities, such as $p(\chi|\mathrm{man})/p(\chi|\mathrm{woman})$ for a word $\chi$, are relevant for discriminating its content. It led to the postulate [27, 30] that for arbitrary words $\chi$ in the corpus,

$$p(\chi|\mathrm{king})/p(\chi|\mathrm{queen}) \approx p(\chi|\mathrm{man})/p(\chi|\mathrm{woman}). \tag{2}$$

If this approximation is an equality, and if the embedding is built from the full rank of the PMI, then Eq.1 can be derived [30]. Yet, how good of an approximation Eq.2 and how large the embedding dimension should be for this argument to work? Will some analogies be learned before others?

To make progress on these questions, further assumptions have been proposed, in particular the existence of a 'true' Euclidean semantic space with some dimension $d$ in which words are associated

with latent vectors. In that space, text generation corresponds to a random walk, such that closer words co-occur more often [25, 27, 29]. Using the assumption that this space is perfectly spherically symmetric, [27] deduces that $W_i \cdot W_j = PMI(i,j)$, and proves that deviations from Eq.2 are tamed, such that Eq.1 holds.

We argue that such views are problematic for two reasons. The proposed symmetry of word embeddings would imply that the spectrum of the PMI corresponds to $d$ identical eigenvalues, whereas in fact the spectrum of the PMI is broadly distributed, as we shall recall. More fundamentally, the assumption that the true Euclidean semantic space has some geometry unconstrained by relations of the kind of Eq.2 appears unlikely. It is inconsistent with the observation that the top eigenvectors of the PMI have semantic content that correlates with analogy relationships [6].

Instead, we base our approach on human psychology experiments describing how words can be characterized by a list of features or attributes [31, 32]. We propose that the co-occurrence of two words is governed by the relationship between their two lists. In that view and in contrast to previous approaches, the semantic space corresponds to the vector of attributes, and is thus organized by relations of the kind of Eq.2. In a simplified setting of binary attributes, the geometry of the semantic space is that of an hypercube (as already mentioned in [31]), yet co-occurrence does not simply depend on the Euclidean distance between word representations. As we shall see, this view naturally explains points (i-iv) above.

## 4. A model for words co-occurrence

Our main assumption is that the occurrence statistics of a word $i$ is governed by the set of attributes that define it [32], such as feminine *v.s.* masculine, royal *v.s.* non-royal, adjective *v.s.* noun, etc. For simplicity, we consider that there are $d$ binary attributes (extending the model to attributes with more than two choices is straightforward, as discussed in Appendix A3). The word $i$ is thus represented as a vector $\boldsymbol{\alpha}_i \in \{-1, +1\}^d$ of the $d$-dimensional hypercube. As a first step, we assume that all the $2^d$ possible words exist. We will see below that even if the vocabulary is much smaller than the total number of possible words, our conclusions hold.

We further assume that different attributes affect the statistics of words in an independent manner (this assumption is relaxed in Appendix A5). As a result, the probability $P(i)$ of word $i$ follows from the set $\{p_k \leq 1/2, k = 1...d\}$, where $p_k$ indicates the probability that attribute $k$ is $+1$:

$$P(i) = \prod_{k=1}^{d} \left( p_k \delta(\alpha_i^{(k)}, 1) + (1 - p_k)\delta(\alpha_i^{(k)}, -1) \right)$$

where $\delta(i,j)$ denotes the Kronecker delta ($\delta(i,j) = 1$ iff $i = j$ and zero otherwise), and $\alpha_i^{(k)}$ denotes the $k^{\text{th}}$ entry of $\boldsymbol{\alpha}_i$. Likewise, the probability $P(i,j)$ that words $i$ and $j$ co-occur follows from the probability $P^{(k)}(\alpha_i^{(k)}, \alpha_j^{(k)})$ that two words with given attributes $\alpha_i^{(k)}$ and $\alpha_j^{(k)}$ co-occur:

$$P(i,j) = P(i)P(j) \prod_{k=1}^{d} P^{(k)}(\alpha_i^{(k)}, \alpha_j^{(k)}) \tag{3}$$

The symmetric matrices $P^{(k)} \in \mathbb{R}^{2 \times 2}$ must be such that $\sum_j P(i,j) = P(i)$, imposing that they can be parametrized by a single scalar $s_k$:

$$P^{(k)} = \begin{pmatrix} 1 + s_k & 1 - q_k s_k \\ 1 - q_k s_k & 1 + q_k^2 s_k \end{pmatrix} \tag{4}$$

where $s_k \in [0, 1]$ characterizes the strength of the "signal" associated to an attribute, and $q_k \equiv p_k/(1 - p_k) \leq 1$ captures the asymmetry in incidence between positive and negative instances of the attribute.

Note that in this noiseless version of the model, Eq.2 holds exactly. Indeed, if $i$ and $i^*$ differ by a single attribute (say the first one) such that $\alpha_i^1 = -\alpha_{i^*}^1 = 1$, and $a$ is any word, then $P(a|i)/P(a|i^*) = P^1(\alpha_a^1, +1)/P^1(\alpha_a^1, -1)$, a result which does not depend on the choice of $i$. Below we will add noise to the model to study how linear analogies persist.

## 5. Word embedding directly from the co-occurrence matrix

Denote by $W$ the $K \times 2^d$ matrix of word embeddings of dimension $K$, whose columns $W_i \in \mathbb{R}^K$ correspond to the embedding of word $i$. Embeddings can be constructed by demanding that the rescaled co-occurrence $M(i,j) \equiv \frac{P(i,j)}{P(i)P(j)}$ is approximated by $W_i \cdot W_j$, or equivalently $M \approx W^T W$. In an $L_2$ sense, the optimal embedding corresponds to:

$$W_i = \sum_{S=1...K} \sqrt{\lambda_S} \cdot v_S(i) u_S \tag{5}$$

where $S$ is the rank of the eigenvector $v_S$ of eigenvalue $\lambda_S$, such that $\lambda_1 \geq \lambda_2 \geq ...\lambda_S$, and the $\{u_S, S = 1...K\}$ is any orthonormal basis. Thus, solving for the word embeddings corresponds to diagonalizing $M$, as we now proceed.

**Theorem**: The matrix $M(i,j) = P(i,j)/(P(i)P(j))$ defined by Eq.3 has eigenvectors

$$v_S = v_{a_1}^{(1)} \otimes v_{a_2}^{(2)} \otimes \cdots \otimes v_{a_d}^{(d)} \quad \text{with } a_k \in \{+, -\} \tag{6}$$

where $\otimes$ indicates a Kronecker product. Its component for word $i$ of attributes $\boldsymbol{\alpha}_i$ is thus

$$v_S(i) \propto v_{a_1}^{(1)}(\alpha_i^1) v_{a_2}^{(2)}(\alpha_i^2) \cdots v_{a_d}^{(d)}(\alpha_i^d). \tag{7}$$

Its associated eigenvalue follows:

$$\lambda_S = \prod_{k=1}^{d} \lambda_{a_k}^{(k)} \tag{8}$$

where the $\lambda_{\pm}^{(k)}$ are eigenvalues of the $2 \times 2$ matrices $P^{(k)}$ defined in Eq.4, and $v_{\pm}^{(k)}$ are the two eigenvectors of these matrices.

**Proof sketch**: This theorem follows from the fact that the matrix $M$ is a Kronecker product of the $d$ matrices $P^{(k)}$, and from standard results on the eigen-decomposition of these products [33]. The block structure of $M$ defining a Kronecker product is most apparent if we order each word $i$ based on the lexicographic order of its attributes $\boldsymbol{\alpha}_i$. That is, the first coordinate that differs between $\boldsymbol{\alpha}_i$ and $\boldsymbol{\alpha}_j$ will determine which word appears first, with $i$ appearing first if its coordinate is $+1$. See Appendix A1 for the detailed proof, a review of Kronecker products, and an explicit construction for the case $d = 2$.

**Diagonalization of the matrices** $P^{(k)}$: This symmetric matrix is diagonalizable, with eigenvalues reported in Appendix A. To study the regime of weak semantic signal, we expand the eigenvalues and eigenvectors to first order in $s_k$. Let $s := s_k$ and $q := q_k$ for brevity. We find in that regime:

$$\lambda_+^{(k)} = 2 + s(1-q)^2/2 + \mathcal{O}(s^2), \quad \lambda_-^{(k)} = s(1+q)^2/2 + \mathcal{O}(s^2) \tag{9}$$

associated to unnormalized eigenvectors $v_+^{(k)}$ and $v_-^{(k)}$ :

$$v_+^{(k)} = \frac{1}{\sqrt{2}} \begin{pmatrix} 1 \\ 1 \end{pmatrix} + \frac{1}{\sqrt{2}} s(q^2-1)/4 \begin{pmatrix} 1 \\ -1 \end{pmatrix} + \mathcal{O}(s^2), \quad v_-^{(k)} = \frac{1}{\sqrt{2}} \begin{pmatrix} -1 \\ 1 \end{pmatrix} + \frac{1}{\sqrt{2}} s(q^2-1)/4 \begin{pmatrix} 1 \\ 1 \end{pmatrix} + \mathcal{O}(s^2) \tag{10}$$

Note that when $q = 1$, i.e., $p_k = \frac{1}{2}$, these vectors reduce for all $s$ to $(1,1)^T$ and $(1,-1)^T$, with the exact eigenvalues $2$ and $2s$.

**Eigensystem of** $M$: The spectrum of $M$ can be obtained using Equation (9) in Equation (8) and the corresponding eigenvectors by using Equation (10) in Equation (7). In the limit where the signals $\{s_k, k = 1...d\}$ are small, the top eigenbands are:

- A top eigenvalue $\lambda_0 = 2^d$ and corresponding eigenvector $\mathbf{1}$ (with higher order corrections linear in the $\{s_k, k = 1...d\}$), corresponding to all $a_k = +$.

- $d$ 'attribute' eigenvectors $v_k$ with $a_{k' \neq k} = +$ and $a_k = -$, giving eigenvalue $\approx 2^{-(d-2)} s_k (1 + q_k)^2$ and implying $v_k(i) \propto \alpha_i^k$.

- $\binom{d}{2} \sim d^2$ eigenvectors of eigenvalues $\propto s_k s_{k'}$, of order $\binom{d}{3} \sim d^3$ eigenvectors of order $\propto s_k s_{k'} s_{k''}$, and so on. At dominant order, the corresponding eigenvectors encode the product of several attributes.

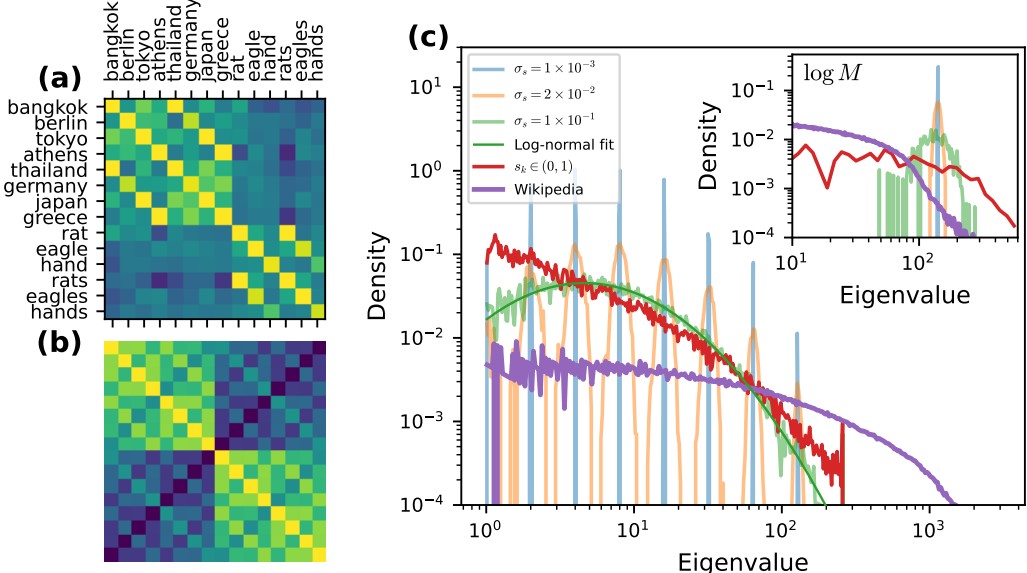

Figure 1: **(a)** A subset of the co-occurrence matrix for Wikipedia data, with labels drawn from the categories "country–capitals" and "noun–plural". **(b)** A subset of the co-occurrence matrix $M$ generated by our model ($d = 8$, $s_k \sim \mathcal{N}(1/2, \sigma_s)$ with $\sigma_s = 10^{-3}$). Colors indicate value of $M_{ij}$ on a log-scale. **(c)** Averaged eigenvalue spectrum of the co-occurrence matrix $M$ for $d = 8$, obtained from 50 random realizations of the semantic strength for $\sigma_s = 10^{-3}$, $2 \times 10^{-2}$, and $10^{-1}$, the uniform $s_k \in (0, 1)$, and empirical co-occurrence data. The inset reveals that the spectrum of the PMI is not peaked with a density of nearly identical eigenvalues, as assumed in some previous works.

As a result, the spectrum of $M$ depends on the distribution of the $s_k$. A representative $M$ with minimal variance in that distribution is shown in Figure 1**b**, in the symmetric case where $q_k = 1$ for all $k$, and can be contrasted with the empirical Wikipedia derived co-occurrences reported in Figure 1**a**. Figure 1**c** shows the density of eigenvalues, averaged over 50 realizations. When the variance in the distribution in the $s_k$'s is small, the eigenvalue spectrum is resolved into discrete bands, while at higher values of the variance in the $s_k$, these bands merge and the distribution is well-fitted by a log-normal distribution.

**Emergence of linear analogy:** Nearly perfect linear analogies appear under the restrictive conditions that (i) the $\{s_k, k = 1...d\}$ are small, (ii) they are narrowly distributed and (iii) the dimension of the word embedding satisfies $K \leq d + 1$. Indeed in that case, the first $K$ eigenvectors into which words are embedded belong to the span of the $v_k$'s and the constant vector $\mathbf{1}$. Embeddings are thus *affine* in the attributes. It implies that if four words $A, B, C, D$ satisfy:

$$\boldsymbol{\alpha}_D = \boldsymbol{\alpha}_A - \boldsymbol{\alpha}_B + \boldsymbol{\alpha}_C$$

Then it must be that:

$$W_A - W_B + W_C = W_D \tag{11}$$

We next show that this property emerges much more robustly if the matrix of elements $PMI(i, j)$ is considered.

## 6. Word embedding from the pointwise mutual information (PMI) matrix

Successful algorithms such as Glove focus on the PMI matrix, as our model justifies. Given word pairs $i, j$ with semantic vectors $\boldsymbol{\alpha}_i, \boldsymbol{\alpha}_j \in \{-1, +1\}^d$, the log co-occurrence matrix or PMI is:

$$\log M(i, j) = \sum_{k=1}^{d} \log P^{(k)}(\alpha_i^{(k)}, \alpha_j^{(k)}).$$

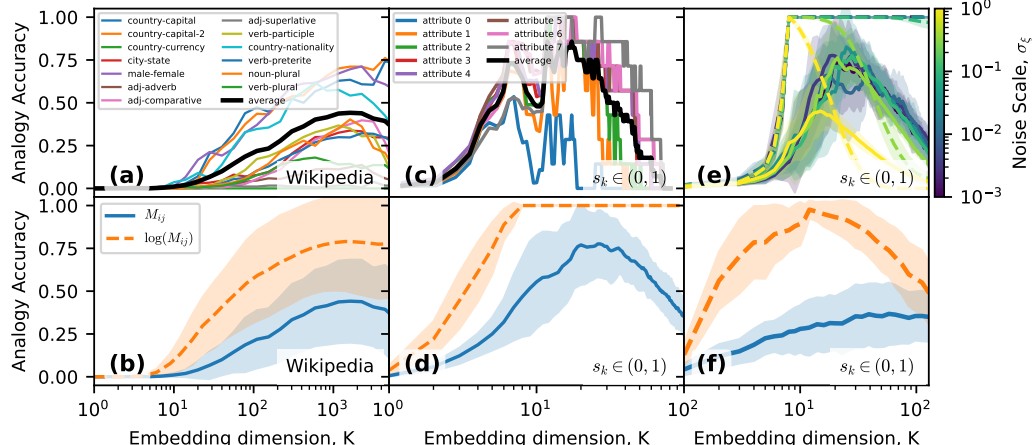

Figure 2: **Analogy completion accuracy emerges at low embedding dimension. (a)**: Wikipedia analogy completion accuracy by analogy category for representations constructed from co-occurrences $M_{ij}$ with a vocabulary of 10,000 words. **(b)**: Analogy accuracy for Wikipedia text, with different matrix targets $M_{ij}$ and $\log(M_{ij} + \varepsilon_R)$ (regularizer $\varepsilon = 10^{-2}$). Shaded area indicates the sample standard deviation across analogy categories. **(c)**: Analogy completion accuracy for a single realization of the model for $d = 8$, for matrix target $M_{ij}$. **(d)**: Analogy accuracy for the model with different matrix targets, averaged across all analogy tasks and 50 realizations of the $s_k$. Shading indicates standard deviation between realizations. **(e)**: Analogy accuracy under the introduction of a multiplicative noise to each entry of the co-occurrence matrix with $P_{ij} \to P_{ij} \exp(\xi_{ij})$ for symmetric $\xi_{ij} \sim \mathcal{N}(0, \sigma_\xi)$ averaged over 10 realizations of $s_k$ and noise $\xi$. **(f)**: Analogy accuracy after both sparsifying the vocabulary and including a multiplicative noise ($\sigma_\xi = 10^{-1}$), retaining only a fraction $f = 0.15$ of words in $d = 12$.

For each attribute $k$, the scalar $\log P^{(k)}(a, b)$, with $a, b \in \{-1, +1\}$, can be written as a bilinear expansion:

$$\log P^{(k)}(a, b) = \delta_k + \eta_k(a + b) + \gamma_k ab,$$

where the values of these coefficients are indicated in Appendix A.

The total matrix takes the form $\log M(i, j) = \delta + \boldsymbol{\eta}^\top \boldsymbol{\alpha}_i + \boldsymbol{\eta}^\top \boldsymbol{\alpha}_j + \boldsymbol{\alpha}_i^\top D \boldsymbol{\alpha}_j$, where $\delta = \sum_k \delta_k$, $\boldsymbol{\eta} = (\eta_1, \ldots, \eta_d)$, and $D = \mathrm{diag}(\gamma_1, \ldots, \gamma_d)$.

**Matrix Form, Rank and Eigenvectors:** Let $A \in \mathbb{R}^{2^d \times d}$ be the matrix with components $A(i, k) = \alpha_i^k$. Then:

$$\log M = \delta \cdot \mathbf{1}\mathbf{1}^\top + ADA^\top + A\boldsymbol{\eta}\mathbf{1}^\top + \mathbf{1}\boldsymbol{\eta}^\top A^\top. \tag{12}$$

**Result 1:** The row space of $A$ is of dimension $d$, thus $\mathrm{rank}(\log M) \leq d + 1$.

**Result 2:** Note that the eigenvectors

$$v_0 := \frac{\mathbf{1}}{\sqrt{2^d}}, \quad v_k := \frac{A_{:,k}}{\sqrt{2^d}} \quad \text{for } k = 1, \ldots, d$$

span the row space of $(\log M)$. The vector $v_k$, previously introduced, simply indicates the value of attribute $k$ for any words. Thus eigenvectors must be affine functions of the attributes.

**Result 3:** Consequently, *the analogy relation of Eq.11 holds exactly in this model, independently of the magnitude or distribution of the $\{s_k, k = 1...d\}$.*

**Result 4:** The eigenvalues of PMI are $\mathcal{O}(2^d)$. It follows from the fact that $v_k^T (\log M) v_k = \mathcal{O}(2^d)$ for $k = 0, ..., d$. When $\boldsymbol{\eta}$ is small ($\boldsymbol{\eta} = 0$ if $q_k = 1$ for all $k$), the matrix $\log M$ has a simple spectral structure, with a top eigenvalue $\lambda_0 = 2^d \delta$ whose eigenvector is approximately $v_0$; and $d$ attribute eigenvalues $\lambda_k = 2^d \gamma_k$ which correspond approximately to the $v_k$'s. All other eigenvalues are zero.

**Numerical validation:** The inset of Figure 1**c** confirms that the spectrum of the PMI matrix lacks the higher-order modes present in the spectrum of $M_{ij}$. For each realization of the PMI matrix in the symmetric case simulated here, there are exactly $d$ semantically relevant eigenvalues in the spectrum.

Next, we turn to the emergence of linear analogies in our model. We define the **top-1 analogy accuracy** by considering how frequently analogy parallelograms defined by equation 11 are approximately satisfied, as

$$\textbf{top-1 accuracy} = \frac{1}{|\mathcal{T}|} \sum_{(A,B,C,D)\in\mathcal{T}} \delta\left(W_D, \operatorname*{argmin}_{D'\in\mathcal{V}}(|W_A - W_B + W_C - W_{D'}|_2)\right) \qquad (13)$$

where $\mathcal{T}$ is a set of analogies consisting of quadruples of words satisfying A:B::D:C, $\delta(i,j)$ denotes the Kronecker delta ($\delta(i,j) = 1$ iff $i = j$ and zero otherwise), and $\mathcal{V}$ denotes the vocabulary of all possible words. That is, we check how often, for a given analogy family $\mathcal{T}$, the analogies in that family, such as (King, Man, Woman, Queen), are satisfied in embedding space, with $W_{\text{Queen}}$ being the closest vector in the vocabulary $\mathcal{V}$ to $W_{\text{King}} - W_{\text{man}} + W_{\text{woman}}$.

In Figure 2**a** we report analogy accuracy using embeddings derived from Wikipedia text co-occurence matrices. The Mikolov et al. analogy task set consists of 19,544 sets of four words analogies, e.g. "hand:hands::rat:rats", divided among 13 families e.g. adjective-superlative, verb-participle, country-nationality, etc. In Figure 2**b** we report the average performance for embeddings obtained with different $M_{ij}$ and $\log(M_{ij})$. The $\log(M)$ target performs strictly better than the raw $M$ target, and saturates at high embedding dimension.

We compute analogy tasks in the symmetric ($q = 1$, i.e. $p_k = 1/2$) binary semantic model (in $d = 8$) as in the Wikpedia text data, by constructing sets of words $\mathcal{T}(k_1, k_2)$ that differ in two semantic dimensions $k_1$ and $k_2$ and satisfy the parallelogram relation. Each analogy is defined by a base word, $\boldsymbol{\alpha_0} = (\alpha_0^{(k)} = \pm 1)$ with $\alpha_0^{(k_1)} = -1$ and $\alpha_0^{(k_2)} = -1$ fixed. There are $2^{d-2}$ such base words (and therefore analogy tuples) in the vocabulary. The analogy is constructed in the obvious way, defining $\boldsymbol{\alpha_1} = \boldsymbol{\alpha_0} + 2\hat{e}_{k_1}$, $\boldsymbol{\alpha_2} = \boldsymbol{\alpha_0} + 2\hat{e}_{k_2}$ and $\boldsymbol{\alpha_{12}} = \boldsymbol{\alpha_0} + 2\hat{e}_{k_1} + 2\hat{e}_{k_2}$ (where $\hat{e}_k$ denotes the $k^{\text{th}}$ standard basis vectors in semantic space). Denoting $\mathbf{w}_0$ the representation of $\boldsymbol{\alpha_0}$, $\mathbf{w}_1$ the representation of $\boldsymbol{\alpha_1}$, etc., the analogy for base word $\boldsymbol{\alpha_0}$ has a score of 1 if

$$\mathbf{w}_{12} = \operatorname{argmin}_{\mathbf{w}\in W} \|(\mathbf{w}_1 - \mathbf{w}_0 + \mathbf{w}_2) - \mathbf{w}\|_2 \qquad (14)$$

and zero otherwise.

In Figure 2**c** we show the emergence of linear analogical reasoning for a single realization of this model for different fixed $k_1$ attributes, averaged over the $k_2 \neq k_1$. The broadly distributed $s_k$'s give rise to analogy performances that vary considerably with embedding dimension. In Figure 2**d**, we show the performance for the same targets that are used in the text data. Validating our results, the $\log(M)$ target achieves 100% accuracy, regardless of the $s_k$ distribution for $K \geq d$. This is better than the $M_{ij}$ target, because performance there is only good when entire eigenbands of the spectrum are included (as we show in the appendix). For broadly distributed $s_k$ (as occurs in real text data, cf. Figure- 1), the bands mix and linear analogical reasoning is lost at increasing $K$.

## 7. Additive Noise Perturbation to the PMI matrix

We now establish the robustness of the spectral structure of $\log M$ under additive noise. Let us consider a perturbed matrix:

$$\log M'(i, j) = \log M(i, j) + \xi(i, j)$$

where $\xi(i, j)$ are independent, zero-mean random variables with bounded variance $\mathbb{E}[\xi(i,j)^2] = \sigma_\xi^2$, and $\xi(i, j) = \xi(j, i)$ to preserve symmetry. We have: $\log M' = \log M + \Delta$ with $\Delta$ the symmetric noise matrix. The spectral norm of the random matrix $\Delta$ is asymptotically $||\Delta||_2 \sim 2\sigma_\xi\sqrt{2^d}$ (see theorem 2.1 of [34]). At fixed $\sigma_\xi$ in the limit of large $d$, this norm is negligible in comparison with distance between the semantic eigenvalues, which is $\mathcal{O}(2^d/d)$ in the non-degenerate case. We can invoke the eigenvalue stability inequality, $|\lambda_k(\log M') - \lambda_k(\log M)| \leq ||\Delta||_2 \sim 2^{d/2}$ (a consequence of Weyl's inequality, see[35]), to justify that the eigen-decomposition of $M$ is thus not affected in the limit of large $d$. The linear analogies of Eq.11 is thus approximately preserved in that limit.

**Numerical validation**: Analogy performance under this perturbation is shown in Figure-(2**e**). We confirm a strong robustness to noise: even with $\sigma_\xi \approx \mathcal{O}(1)$, excellent analogy accuracy is possible for the PMI case. The performance of linear analogies degrades at high $K$ when enough "bad" eigenvectors from the noise are introduced. This occurs at an embedding dimension of order $K \sim \frac{1}{\sigma_\xi} 2^{d/2}$, as shown in appendix.

## 8. Spectral Stability under Vocabulary Subsampling

Our model assumes that all the possible $2^d$ combinations of attributes are incarnated into existing words, an assumption that is clearly unrealistic. We now show that even if we randomly prune an immense fraction $f$ of the words, the spectral properties of the PMI matrix are remarkably robust, as long as the number of words $m = f2^d \gg d$.

We analyze the impact of this sampling procedure on the eigendecomposition of $\log M$ in the symmetric case $p_k = 1/2$ for all $k$, and focus on the interesting, non-constant part of the PMI matrix. From Eq.12 we obtain:

$$\log M - \delta \cdot \mathbf{1}\mathbf{1}^T = ADA^\top = \tilde{A}\tilde{A}^\top \tag{15}$$

where $\tilde{A} = AD^{1/2}$. After pruning, we restrict $\tilde{A}$ to the retained vocabulary, yielding a matrix $\tilde{A}_S \in \mathbb{R}^{m \times d}$. Our goal is to study the eigen-decomposition of $\tilde{A}_S\tilde{A}_S^\top$. Its positive spectrum is identical to that of the $d \times d$ Gram matrix:

$$G := \tilde{A}_S^\top \tilde{A}_S = D^{1/2}A_S^\top A_S D^{1/2}$$

Let $\boldsymbol{\alpha}_i^T$ denote a row of $A_S$. Each $\boldsymbol{\alpha}_i$ is drawn independently from a uniform distribution on the hypercube. We have $G = D^{1/2} \sum_{i=1\ldots m} \boldsymbol{\alpha}_i \boldsymbol{\alpha}_i^T D^{1/2}$. Thus, $\mathbb{E}[G] = mD^{1/2}\Sigma D^{1/2}$ where $\Sigma_{kl} = \mathbb{E}[\alpha^{(k)}\alpha^{(l)}] = I$.

The convergence of $\Sigma$ to the identity is described by the Marchenko–Pastur theory for sample covariance matrices of the form $\sum_{i=1\ldots m} \boldsymbol{\alpha}_i \boldsymbol{\alpha}_i^T / m$, stating that eigenvalues converge to unity as $m/d \to \infty$ [36]. Thus for $m \gg d$, the eigenvalues of $\log M_S$ follow that of the un-pruned case, except for a trivial rescaling of magnitude $m/2^d$.

Acting with the operator $A_S$ on eigenvectors of the Gram matrix, one obtains the desired eigenvectors of $\tilde{A}\tilde{A}^\top$. As $m/d \to \infty$, the Gram matrix eigenvectors are simply the set $e_k \in \mathbb{R}$ of basis vectors associated with a single attribute, and $A_S e_k \propto v_k$ is the attribute $k$ vector introduced above. The limit $m \gg d$ thus recovers the eigen-decomposition of the un-pruned matrix, and analogy must be recovered.

**Numerical validation**: in Figure- (2**f**), we report results on a sparsified variation of the model in $d = 12$ for retention probability $f = 0.15$ (see Appendix for a representative sparsified co-occurrence matrix). Despite both a multiplicative noise and the removal of $> 97\%$ of the co-occurrence matrix, excellent analogical reasoning performance can be obtained for $K \geq d$.

## 9. Analogy structure after removal of all pairs of a given relationship

Remarkably, it is found that a linear analogical structure persists even when all direct word pairs of a given attribute are removed from the training set [8]. Considering for example the masculine-feminine relationship, even if all sentences where all the pairs of the type (king, queen), (man, woman), (actor, actress), are removed, one still obtains that king-queen+woman=man.

This observation is naturally explained in our model. Fix an attribute index $k = 1$. Consider all the $2^{d-1}$ pairs $(i, j)$ such that words $i$ and $j$ differ *only* in the value of the first attribute $k$, i.e., $\alpha_i^{(l)} = \alpha_j^{(l)}$ for all $l \neq 1$, and $\alpha_i^1 = -\alpha_j^1$. Next, define a modified co-occurrence matrix $\log(M'(i, j))$ where we set $\log(M'(i, j)) = 0$ for these $2^{d-1}$ entries, while leaving all other entries identical to those of $M$. Let $\Delta = M' - M$ be the resulting perturbation.

The operator norm of $\Delta$ is bounded by $\text{trace}(\Delta^T\Delta)^{1/2} \sim \sqrt{2^d}$ (since it is a sparse matrix which consists of $2^d$ elements of order one), which is negligible with respect to the eigenvalues of $M$, of order $2^d$ as discussed in the previous section. Again using the eigenvalue stability inequality, we

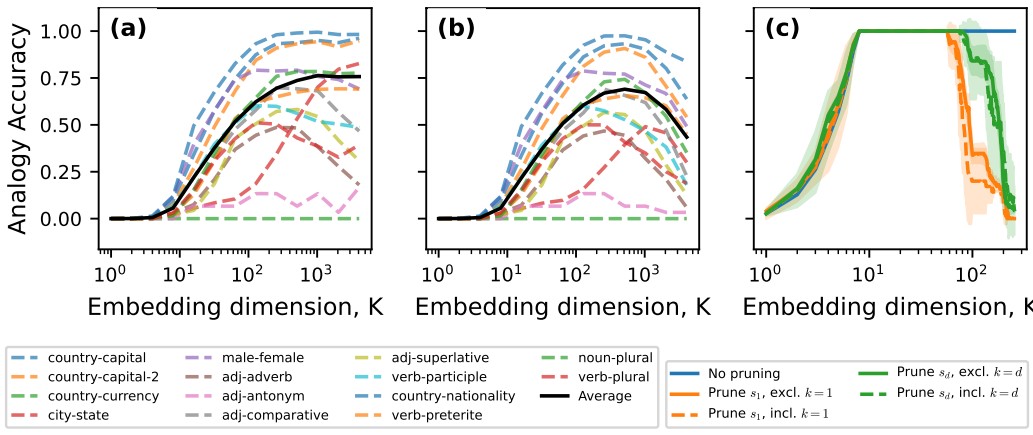

Figure 3: **(a)** Performance of the $\log(M)$ Wikipedia text co-occurrence matrix for analogy tasks. **(b)** As in (a), but having pruned from the co-occurrence matrix all co-occurrences of pairs matching the indicated analogy. The average, in black, reports the analogy accuracy when all analogies are pruned from the corpus. **(c)** Pruning of analogies from the co-occurrence matrix in the symmetric synthetic model over ten realizations of $s_k \in (0,1)$ disorder for $d = 8$. Solid curves represent the average analogy performance on analogies involving the unpruned dimension, while the dashed curve reports performance on analogies that involve the dimension affected by pruning.

obtain that if eigenvalues of $M$ are not degenerate, its eigenvectors and eigenvalues are not affected in the limit of large $d$. Consequently, the embedding of each word is unaffected in this limit, and Eq.11 still holds.

**Numerical validation:** In Figure 3**a** we show the performance of the PMI matrix for different analogy families. For each family, we construct a pruned PMI matrix, with co-occurrences of pairs matching the analogy set to zero, and find a minimal performance degradation in analogy accuracy in Figure 3**b**. We try this experiment in the model, pruning either the strongest semantic direction (WLOG, $k = 1$) or the weakest semantic direction (WLOG) $k = d$ in Figure 3**c**. In both cases, linear analogies survive this perturbation, with perfect accuracy appearing at $K = d$. Analogies that include the pruned dimension perform similarly to analogies that do not include the pruned dimension. As with the real data (Figure 3**b**), this pruning introduces additional noisy eigenvectors to the representation at high embedding dimension which eventually leads to a breakdown of linear analogies.

## 10. Discussion

Word embeddings are central to the interpretation of large language models. Surprisingly, linear subspaces characterizing semantically meaningful concepts, originally found in classical word embedding methods, are also realized in large language models [12, 13, 37, 38] where they can enable control of model behavior [17] and are crucial for fact retrieval [39]. While these linear subspaces are sometimes encoded in a context-dependent manner [15], it is possible to obtain context-free LLM embeddings (as in [40]) to test our views in modern LLMs. The prevalence of these linear subspaces suggest that the statistics of language plays a fundamental role, and revisiting classical word embedding methods allows us to develop a sharper understanding of this question.

We have shown that linear analogies in the word embeddings in such algorithms naturally arise if words are characterized by a list of attributes, and if each attribute affects the context of their associated word in an independent manner. We formulated a simple, analytically tractable model of co-occurrence statistics that captures this view, where attributes are binary and all combinations of attributes correspond to a word. This model rationalizes various observations associated with the emergence of linear analogies, and provides a fine-grained description of how they depend on the embedding dimension and the co-occurrence matrix considered. Remarkably, the model is extremely

robust to perturbations including noise, the sparsification of the word vocabulary, the introduction of correlations between semantic attributes, or the removal of all pairs associated to a specific relation.

### Limitations

Our model is obviously a great simplification of actual word statistics. For example, polysemantic words such as *bank* (a bank, river bank, to bank) complicate co-occurrence statistics. Furthermore, some attributes may be hierarchically organized [38]; this property can be captured by random hierarchy models [41] and is revealed by diffusion models [42, 43]. The possibility that such properties may be studied from co-occurrence alone is an intriguing question for future work. It calls for the development of improved analytically tractable models that capture these effects.

### Data, code availability, and compute budget

The code used to produce the model results, Wikipedia co-occurence statistics, and figures is available on GitHub at `https://github.com/DJKorchinski/linear-analogies-word-embedding-reproduction` and in the supplementary files. All simulations together run in under 150 minutes on an Nvidia H100.

## Acknowledgments and Disclosure of Funding

D. J. Korchinski acknowledges financial support from the Natural Sciences and Engineering Research Council of Canada (NSERC PDF - 587940 - 2024). D. Karkada thanks Google DeepMind and BAIR for funding support. This work was supported by the Simons Foundation through the Simons Collaboration on the Physics of Learning and Neural Computation (Award ID: SFI-MPS-POL-00012574-05), PIs Bahri and Wyart.

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

## Appendices

## A1. Eigenspectrum of the $P^{(k)}$ matrices.

Recalling the definition of the $P^{(k)}$ matrix,

$$P^{(k)} = \begin{pmatrix} 1 + s_k & 1 - q_k s_k \\ 1 - q_k s_k & 1 + q_k^2 s_k \end{pmatrix} \tag{16}$$

the eigenvalues are given by:

$$\lambda_{\pm}^{(k)} = 1 + \frac{s_k}{2}(1 + q_k^2) \pm \frac{1}{2}\sqrt{s_k^2(1 + q_k^2)^2 - 8q_k s_k + 4} \tag{17}$$

and the corresponding eigenvectors are

$$v_{\pm} = \begin{bmatrix} \frac{(1-q^2)s \pm \sqrt{(q^2+1)^2 s^2 - 8qs + 4}}{2(1-qs)} \\ 1 \end{bmatrix}. \tag{18}$$

In the special (symmetric) case that $q = 1$, the eigenvalues are simply 2 and $2s$ for the $-$ and $+$ cases respectively with eigenvectors $v_{\pm} = [1, \pm 1]^{\top}$

### Thereom proof

**Theorem**: The matrix $M(i,j) = P(i,j)/(P(i)P(j)) = \prod_k^d P^{(k)}(\alpha_i^{(k)}, \alpha_j^{(k)})$ indexed by word $i$ of attributes $\boldsymbol{\alpha}_i$ and word $j$ of attributes $\boldsymbol{\alpha}_j$ defined by Equation (16) has eigenvectors

$$v_S = v_{a_1}^{(1)} \otimes v_{a_2}^{(2)} \otimes \cdots \otimes v_{a_d}^{(d)} \quad \text{with } a_k \in \{+1, -1\} \tag{19}$$

where $\otimes$ indicates a Kronecker product. Its component for word $i$ of attributes $\boldsymbol{\alpha}_i$ is thus

$$v_S(i) \propto v_{a_1}^{(1)}(\alpha_i^1)v_{a_2}^{(2)}(\alpha_i^2) \cdots v_{a_d}^{(d)}(\alpha_i^d). \tag{20}$$

with associated eigenvalue:

$$\lambda_S = \prod_{k=1}^{d} \lambda_{a_k}^{(k)} \tag{21}$$

where the $\lambda_{\pm}^{(k)}$ are eigenvalues of the $2 \times 2$ matrices $P^{(k)}$ defined in Equation (16), and $v_{\pm}^{(k)}$ are the two eigenvectors of these matrices given in Equation (18).

*Proof.* This theorem follows from the fact that the matrix $M$ is a Kronecker product of the $d$ matrices $P^{(k)}$, and from standard results on the eigen-decomposition of these products [33]. It suffices to show that $M$ matrix admits a Kronecker representation. We show this by induction. Let

$$M^{(d')}(i, j) = \prod_{k=1}^{d'} P^{(k)}(\alpha_i^{(k)}, \alpha_j^{(k)}) \tag{22}$$

denote scaled co-occurrence matrix up to semantic dimension $d'$. Clearly $M = M^{(d)}$. We will show that $M^{(d')} = \otimes_{k=1}^{d'} P^{(k)}$.

To do so, we consider a canonical basis whose basis vectors correspond to words. Word $i$ is the vector of zero everywhere, except at a position $n(\boldsymbol{\alpha}_i, d)$ that follows:

$$n(\boldsymbol{\alpha}_i, d) = \sum_{k=1}^{d} 2^{d-k} \delta_{\alpha_i^{(k)}, +1}, \tag{23}$$

Note that the binary representation of $n(\boldsymbol{\alpha}_i, d)$ is simply given by $\alpha_i^{(k)}$, with the change $\alpha_i^{(k)} = -1 \to 0$ and $\alpha_i^{(k)} \to 1$. A useful consequence of this definition is that

$$n(\boldsymbol{\alpha}, d+1) = 2n(\boldsymbol{\alpha}, d) + \delta_{\alpha^{d+1}, +1} = 2n(\boldsymbol{\alpha}, d) + n(\alpha^{d+1}, 1). \tag{24}$$

*Base case*: For a single semantic dimension, we trivially have that

$$M^{(d'=1)} = P^{(1)},$$

which is the Kronecker product of one term.

*Induction step*: Assume that $M^{(d')} = \otimes_k^{d'} P^{(k)}$ for some $d' \geq 1$. We must show that $M^{(d'+1)} = M^{(d')} \otimes P^{(d'+1)}$.

By the definition in Equation (22), we have:

$$M^{(d'+1)}(i,j) = \prod_{k=1}^{(d'+1)} P^{(k)}(\alpha_i^{(k)}, \alpha_j^{(k)}) = \left( \prod_k^{d'} P^{(k)}(\alpha_i^{(k)}, \alpha_j^{(k)}) \right) P^{(d'+1)}(\alpha_i^{(d'+1)}, \alpha_j^{(d'+1)})$$

Using the row-indexing notation of Equation (23), this expression can be written as:

$$M^{(d'+1)}_{n(i,d'+1),n(j,d'+1)} = M^{(d')}_{n(\boldsymbol{\alpha}_i,d'),n(\boldsymbol{\alpha}_j,d')} P^{(d'+1)}_{n(\alpha_i^{(d')},1),n(\alpha_j^{(d')},1)}$$

The Kronecker product between an arbitrary matrix $A$ and a matrix $B \in \mathbb{R}^{p \times q}$ is defined as

$$(A \otimes B)_{pi+i',qj+j'} = A_{ij} B_{i'j'}.$$

With $B = P^{(d'+1)} \in \mathbb{R}^{2 \times 2}$, we then have

$$M^{(d'+1)}_{n(i,d'+1),n(j,d'+1)} = (M^{(d')} \otimes P^{(d'+1)})_{2n(\boldsymbol{\alpha}_i,d')+n(\alpha_i^{(d'+1)},1),2n(\boldsymbol{\alpha}_j,d')+n(\alpha_j^{(d'+1)},1)}$$

and using the indexing identity of Equation (24), this simplifies to

$$M^{(d'+1)}_{n(i,d'+1),n(j,d'+1)} = (M^{(d')} \otimes P^{(d'+1)})_{n(\boldsymbol{\alpha}_i,d'+1),n(\boldsymbol{\alpha}_j,d'+1)}$$

Thus

$$\implies M^{(d'+1)} = M^{(d')} \otimes P^{(d'+1)}.$$

By the assumption of our inductive step, we thus get:

$$M^{(d'+1)} = \otimes_{k=1}^{d'+1} P^{(k)}.$$

$\square$

### Properties of Kronecker products

It may be instructive to review a few standard results on the eigen-decomposition Kronecker products. For a product of matrices $\mathbf{A} \in \mathbb{R}^{m \times n}$ with

$$(\mathbf{A} \otimes \mathbf{B}) = \begin{bmatrix} A_{11}\mathbf{B} & \dots & A_{1n}\mathbf{B} \\ \vdots & \ddots & \vdots \\ A_{m1}\mathbf{B} & \dots & A_{mn}\mathbf{B} \end{bmatrix} \tag{25}$$

if $\lambda_u$ and $\mathbf{u}$ are a eigen(value/vector) of $\mathbf{A}$ so $\lambda_u \mathbf{u} = \mathbf{A}\mathbf{u}$ and $\lambda_v$ and $\mathbf{v}$ similarly satisfy $\lambda_v \mathbf{v} = \mathbf{B}\mathbf{v}$, then the vector

$$\mathbf{u} \otimes \mathbf{v} = \begin{bmatrix} u_1 \mathbf{v} \\ u_2 \mathbf{v} \\ \vdots \\ u_n \mathbf{v} \end{bmatrix} \tag{26}$$

is an eigenvector of $\mathbf{A} \otimes \mathbf{B}$ with eigenvalue $\lambda_u \lambda_v$ [33]. This can be seen schematically, with:

$$(\mathbf{A} \otimes \mathbf{B})(\mathbf{u} \otimes \mathbf{v}) = \begin{bmatrix} A_{11}u_1\mathbf{B}\mathbf{v} & + & \dots & + & A_{1n}u_n\mathbf{B}\mathbf{v} \\ \vdots & & \ddots & & \vdots \\ A_{m1}u_1\mathbf{B}\mathbf{v} & + & \dots & + & A_{mn}u_n\mathbf{B}\mathbf{v} \end{bmatrix}$$

$$= \begin{bmatrix} \lambda_v \mathbf{v} \sum_i A_{1i}u_i \\ \vdots \\ \lambda_v \mathbf{v} \sum_i A_{mi}u_i \end{bmatrix}$$

$$= \lambda_v \lambda_u \begin{bmatrix} u_1 \mathbf{v} \\ \vdots \\ u_n \mathbf{v} \end{bmatrix}$$

$$= \lambda_v \lambda_u (\mathbf{u} \otimes \mathbf{v})$$

**Explicit construction for d=2**

In this section, we provide an explicit example of the construction of the $P^{(k)}$ matrices and the eigenvectors and eigenvalues for the case $d = 2$, which is the smallest case to support linear analogy.

Consider the symmetric $q = 1$ case for $d = 2$ with two semantic strengths $s_1$ and $s_2$. The $P^{(k)}$ matrices are:

$$\mathbf{P}^{(1)} = \begin{pmatrix} s_1 + 1 & 1 - s_1 \\ 1 - s_1 & s_1 + 1 \end{pmatrix} \tag{27}$$

and

$$\mathbf{P}^{(2)} = \begin{pmatrix} s_2 + 1 & 1 - s_2 \\ 1 - s_2 & s_2 + 1 \end{pmatrix} \tag{28}$$

with eigenvectors $v_{\pm} = (1, \pm 1)^\top$ and eigenvalues $\lambda_{\pm} = 2s^{(1 \mp 1)/2}$. The matrix $\mathbf{M}$ is given by:

$$\mathbf{M} = \begin{bmatrix} (1 + s_1)\mathbf{P}^{(2)} & (1 - s_1)\mathbf{P}^{(2)} \\ (1 - s_1)\mathbf{P}^{(2)} & (1 + s_1)\mathbf{P}^{(2)} \end{bmatrix}$$

$$= \begin{bmatrix} (1 + s_1)(1 + s_2) & (1 + s_1)(1 - s_2) & (1 - s_1)(1 + s_2) & (1 - s_1)(1 - s_2) \\ (1 + s_1)(1 - s_2) & (1 + s_1)(1 + s_2) & (1 - s_1)(1 - s_2) & (1 - s_1)(1 + s_2) \\ (1 - s_1)(1 + s_2) & (1 - s_1)(1 - s_2) & (1 + s_1)(1 + s_2) & (1 + s_1)(1 - s_2) \\ (1 - s_1)(1 - s_2) & (1 - s_1)(1 + s_2) & (1 + s_1)(1 - s_2) & (1 + s_1)(1 + s_2) \end{bmatrix}$$

The eigenvalues are $\lambda_0 = 2^2$, $\lambda_1 = 2^2 s_1$, $\lambda_2 = 2^2 s_2$, $\lambda_{12} = 2^2 s_1 s_2$ with corresponding eigenvectors: $v_0 = (1, 1, 1, 1)^\top$, $v_1 = (-1, -1, 1, 1)^\top$, $v_2 = (1, -1, 1, -1)^\top$, and $v_{12} = (1, -1, -1, 1)^\top$. These are precisely the eigenvectors and eigenvalues guaranteed by the theorem.

In $d = 2$, there is exactly one analogy possible. It can of course be made explicit, by using the following mapping of words to semantic attributes

$$\boldsymbol{\alpha}_{\text{Man}} = (-1, -1)$$
$$\boldsymbol{\alpha}_{\text{Woman}} = (-1, 1)$$
$$\boldsymbol{\alpha}_{\text{King}} = (1, -1)$$
$$\boldsymbol{\alpha}_{\text{Queen}} = (1, 1)$$

where the attribute $s_1$ corresponds to "Royalty" and $s_2$ to "Gender". The word embeddings for $K = 2$ and $K = 3$ are given in Table 1. These embeddings are generated by using the embedding matrix $\mathbf{W}_2 = [\mathbf{v}_0, \mathbf{v}_1]^\top$ and $\mathbf{W}_3 = [\mathbf{v}_0, \mathbf{v}_1, \mathbf{v}_2]^\top$ for $K = 2$ and $K = 3$ respectively. The representation of each word is the corresponding column of $\text{diag}(\lambda_0, \lambda_1, \lambda_2)\mathbf{W}$. For example: $w_{\text{Queen},i} = [\text{diag}(\sqrt{\lambda_0}, \sqrt{\lambda_1}, \sqrt{\lambda_2})\mathbf{W}]_{i,\text{Queen}}$.

The only possible analogy (up to trivial permutations) in $d = 2$ is then given by the equation:

$$\boldsymbol{w}_{\text{Queen}} - \boldsymbol{w}_{\text{Woman}} + \boldsymbol{w}_{\text{Man}} = \boldsymbol{w}_{\text{King}} \tag{29}$$

which is satisfied uniquely when $K = 3$, but not when $K = 2$ as is depicted in fig. 4 and fig. 5. As the example here shows, each successive eigenvector (after the first trivial $\mathbf{1}$ vector) included in the

| Word | $K = 2$ Embedding | $K = 3$ Embedding |
|------|-------------------|-------------------|
| Man | $2(1, -\sqrt{s_1})$ | $2(1, -\sqrt{s_1}, -\sqrt{s_2})$ |
| Woman | $2(1, -\sqrt{s_1})$ | $2(1, -\sqrt{s_1}, +\sqrt{s_2})$ |
| King | $2(1, +\sqrt{s_1})$ | $2(1, +\sqrt{s_1}, -\sqrt{s_2})$ |
| Queen | $2(1, +\sqrt{s_1})$ | $2(1, +\sqrt{s_1}, +\sqrt{s_2})$ |

Table 1: The explicitly constructed embeddings for the words "Man", "Woman", "King" and "Queen" in the $d = 2$ case. The $K = 3$ embeddings are obtained by adding a third dimension corresponding to the semantic strength $s_2$.

$$w_{\text{Woman}} = w_{\text{Man}} = 2\left(1, -\sqrt{s_1}\right) \qquad \boldsymbol{w}_{\text{Man}} - \boldsymbol{w}_{\text{Woman}}$$

$$O\, w_{\text{Queen}} = w_{\text{King}} = 2\left(1, \sqrt{s_1}\right)$$

Figure 4: A degenerate analogy when $K = 2$, since both $\boldsymbol{w}_{\text{Queen}}$ and $\boldsymbol{w}_{\text{King}}$ satisfy $x = \boldsymbol{w}_{\text{Queen}} + (\boldsymbol{w}_{\text{Man}} - \boldsymbol{w}_{\text{Woman}})$

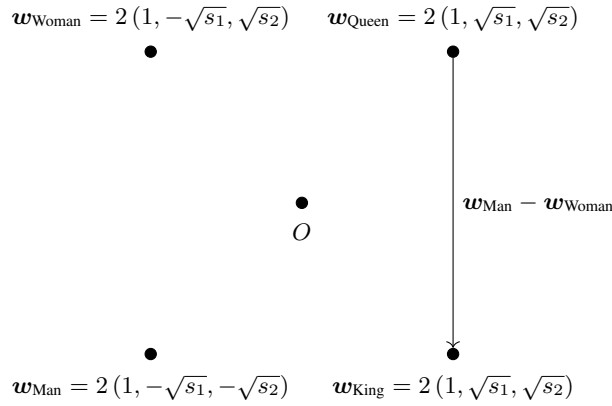

Figure 5: The analogy is no-longer degenerate when $K = 3$.

representation allows the embedding to disambiguate one additional semantic direction – with $K = 2$ only the royalty axis present in the representation, while in $K = 3$, both the royalty and gender axes are present.

## A2. Coefficients characterizing the PMI matrix

The definitions for the three parameters $\delta_k$, $\eta_k$ and $\gamma_k$ are as follows:

$$\delta_k = \frac{1}{4}\left[\log(1 + s_k) + \log(1 + q_k^2 s_k) + 2\log(1 - q_k s_k)\right],$$

$$\eta_k = \frac{1}{4}\left[\log(1 + s_k) - \log(1 + q_k^2 s_k)\right],$$

$$\gamma_k = \frac{1}{4}\left[\log(1 + s_k) + \log(1 + q_k^2 s_k) - 2\log(1 - q_k s_k)\right].$$

## A3. Tertiary model with neutral attribute

We now generalize our model to the case where each semantic attribute $\alpha^{(k)}$ can take one of three values: $-1$, $0$, or $+1$. The value $0$ corresponds to a *neutral* setting, indicating that the word does not express this semantic dimension. This extension allows us to model richer vocabularies where many attributes may be inactive for a given word. For simplicity of notations, we consider the case where the three values of each attribute are equally likely.

We define the co-occurrence matrix $P^{(k)}$ for attribute $k$ as a symmetric $3 \times 3$ matrix, where the rows and columns correspond to $\{-1, +1, 0\}$. We assume:

$$P^{(k)} = \begin{pmatrix} 1 + s_k + \frac{f_k}{2} & 1 - s_k + \frac{f_k}{2} & 1 - f_k \\ 1 - s_k + \frac{f_k}{2} & 1 + s_k + \frac{f_k}{2} & 1 - f_k \\ 1 - f_k & 1 - f_k & 1 + 2f_k \end{pmatrix}$$

Here: - $s_k \in [0, 1]$ encodes the strength of similarity between matching vs. opposite polarities ($+1$ vs. $-1$), - $f_k \ll 1$ is the frequency of non-neutral settings for attribute $k$.

This matrix can be decomposed as:

$$P^{(k)} = A^{(k)} + f_k B^{(k)} + \mathcal{O}(f_k^2)$$

where the unperturbed component is:

$$A^{(k)} = \begin{pmatrix} 1 + s_k & 1 - s_k & 1 \\ 1 - s_k & 1 + s_k & 1 \\ 1 & 1 & 1 \end{pmatrix}, \quad B^{(k)} = \begin{pmatrix} \frac{1}{2} & \frac{1}{2} & -1 \\ \frac{1}{2} & \frac{1}{2} & -1 \\ -1 & -1 & 2 \end{pmatrix}$$

We now analyze the eigenvalues of $P^{(k)}$ using first-order perturbation theory in $f_k$.

**Eigenvectors of $A^{(k)}$.** This symmetric matrix has three orthonormal eigenvectors:

$$v_0 = \frac{1}{\sqrt{3}}(1, 1, 1)^\top \qquad \text{(constant mode)}$$

$$v_1 = \frac{1}{\sqrt{2}}(1, -1, 0)^\top \qquad \text{(contrast between } -1 \text{ and } +1)$$

$$v_2 = \frac{1}{\sqrt{6}}(1, 1, -2)^\top \qquad \text{(contrast between neutral and polar values)}$$

**Resulting eigenvalues of $P^{(k)}$.** Up to first order in $f_k$, perturbation theory gives:

$$\lambda_0^{(k)} = 3 + \mathcal{O}(f_k^2)$$
$$\lambda_1^{(k)} = 2s_k + \mathcal{O}(f_k^2)$$
$$\lambda_2^{(k)} = 6f_k + \mathcal{O}(f_k^2)$$

**Interpretation.** - The top eigenvalue corresponds to the constant mode and is unaffected at linear order in $f_k$,

- The contrast direction between $-1$ and $+1$ preserves the same eigenvalue as in the binary case: $2s_k$,

- A new third direction, orthogonal to both, emerges with small but nonzero eigenvalue $6f_k$, reflecting the semantic impact of neutral attribute values.

This shows that even in the presence of neutral words, the dominant structure of the embedding space — necessary for analogy — remains intact up to small corrections.

## A4. Additional numerical study of the model

Here, we report several additional experiments on the symmetric binary semantics model.

**Dependence of analogy accuracy on disribution of the $s_k$**

In Figure 6, we measure the analogy accuracy for narrowly distributed semantic strengths $s_k \sim \mathcal{N}(1/2, 10^{-3})$. Accuracy with the $M_{ij}$ matrix target is perfect whenever a complete eigenband is included in the representation. Perfect analogy reconstruction is possible for the symmetric model, but is very fragile and requires a narrow distribution of $s_k$.

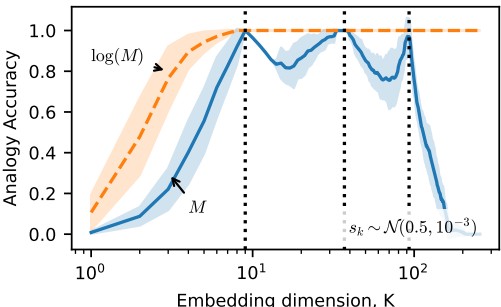

Figure 6: Analogy performance for narrowly distributed $s_k$ for different matrix targets in $d = 8$. Shading represents the standard deviation across 50 replicates. Vertical lines at $K_1 = 1 + \binom{d}{1}$, $K_2 = K_1 + \binom{d}{2}$ and $K_3 = K_2 + \binom{d}{3}$ mark the complete inclusion of the different eigenbands.

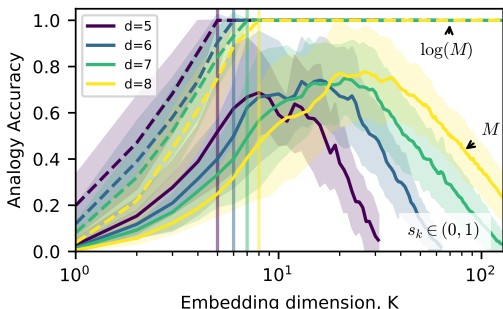

Figure 7: Analogy performance for uniformly distributed $s_k$ in different dimensions. Shading represents the standard deviation across 50 replicates. Vertical lines mark $K = 5$, $K = 6$, $K = 7$, and $K = 8$, corresponding to the dimension of the semantic embedding space for the main curves.

When there is greater variance in the $s_k$ distribution, e.g. $s_k \in (0, 1)$, then the higher-order eigenvalues (corresponding to $s_k s_{k'}$) begin to be included, without first capturing all of the first-order eigenvalues. In this case, perfect analogy accuracy is typically not attained and the peak in accuracy happens at higher $d$, as can be seen in Figure 7. In the $\log(M)$ case, the semantic eigenvalues are all included at $K = d$, as is indicated by the vertical lines in Figure 7.

**Vocabulary sparsification**

In Figure 8, we show the effect of sparsification on the co-occurrence matrix $M$ at a fraction $f = 0.15$. Each row $i$ in $M$ corresponds to co-occurrences including word $i$. By removing row $i$ and column $i$, we effectively remove word $i$ from the vocabulary. Despite removing $\approx 98\%$ of the co-occurrence matrix, the overall hierarchical structure is still readily apparent. We test the effect of sparsification on the top $d$ semantic eigenvalues of a $\log(M)$ target in Figure 9. As is predicted by the theory, the top $d$ eigenvalues are minimally perturbed by sparsification beyond a simple rescaling of $f = m/2^d$. This breaks down only when the number of retained words approaches $m \to d$, meaning that a tremendously sparsified vocabulary still retains the same eigenvalue structure.

**Breakdown of analogy accuracy with increasing $K$ in the presence of noise.**

Here, we study the effect of introducing a noise perturbation $\log(M'_{ij}) = \log(M_{ij}) + \xi_{ij}$ onto the PMI matrix, with $\xi_{ij} \sim \mathcal{N}(0, \sigma_\xi)$. The spectral density has two prominent features: a set of "semantic" eigenvalues, with $\lambda_k \sim 2^d \gamma_k$ and a set of small, noisy eigenvalues starting at a scale set by $\sigma_s$, as can be seen in Figure 10**a**.

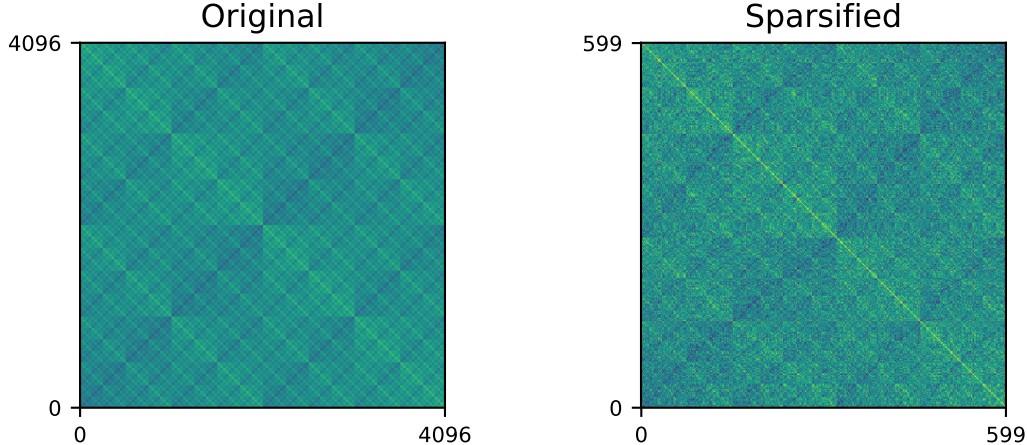

Figure 8: The effect of sparsification ($f = 0.15$) on a realization of the co-occurrence matrix with $d = 12$ and $s_k \in (0, 1)$. Colours represent the value of $\log(M)$.

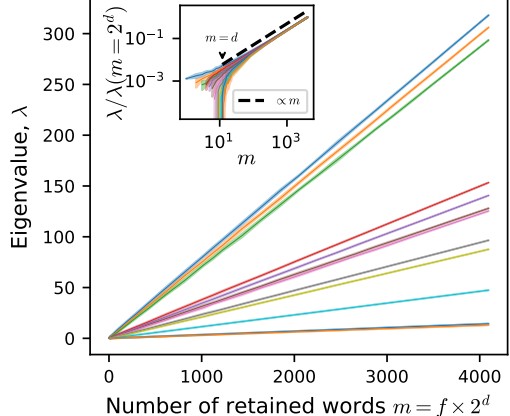

Figure 9: The value of the top $d = 12$ eigenvalues for a $\log(M)$ matrix with $s_k \in (0, 1)$ as a function of sparsification. Shaded area reflects the standard deviation across 20 realizations of sparsification, for the same fixed $s_k$. Inset is the same data, but rescaling by the asymptotic value of each eigenvalue so as to effect a collapse for $m > d$.

The $2^d \times 2^d$ dimensional $\boldsymbol{\xi}$ matrix has a spectral norm converging to $||\boldsymbol{\xi}|| \sim 2\sigma_s\sqrt{2^d}$ as $d \to \infty$. This scale collapses the noise floor of $\log(M')$ across both dimension and $\sigma_s$ as can be seen in Figure 10**c**.

As the the embedding dimension is increased, additional noisy eigenvectors are included in the representation. The $\tilde{\boldsymbol{w}}$ representation vectors are constructed from the $\boldsymbol{w}$ eigenvectors as

$$\tilde{\boldsymbol{w}}_k \equiv \sqrt{\lambda_k}\boldsymbol{w} . \tag{30}$$

We expect these noisy eigenvectors interfere with analogical reasoning when their combined magnitude is of the same order of magnitude as the semantic eigenvectors, i.e. when

$$\left| \sum_{k=d+1}^{K} \tilde{\boldsymbol{w}}_k \right|_2 \approx |\tilde{\boldsymbol{w}}_1|_2 \sim 2^d .$$

Since the $\tilde{\boldsymbol{w}}_{k>d}$ are orthogonal, in the limit $K \gg d$, we have that $\left| \sum_{k=d+1}^{K} \tilde{\boldsymbol{w}}_k \right|_2 \approx \sqrt{K}\sigma_s 2^{d/2}$. We expect linear analogies to break down when we have an embedding dimension of order

$$\implies K \approx \frac{1}{\sigma_s}2^{d/2} .$$

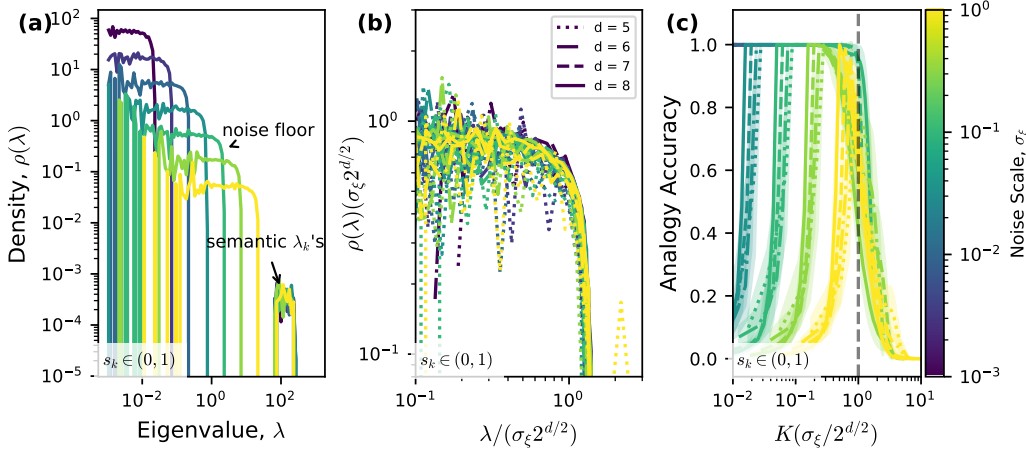

Figure 10: **(a)** Eigenvalue spectrum of the PMI matrix with the addition of an elementwise independently and identically distributed gaussian noise, for $d = 8$ and $N = 10$ replicates. **(b)** As in **a** rescaled to collapse the noise floor across both noise scale and dimension. **(c)** The accuracy of analogy tasks for different dimensions and noise scales, rescaling the embedding dimension $K$ to collapse the breakdown in analogy accuracy at high $d$.

This is confirmed by the collapse in Figure 10**c**.

## A5. Extension with correlated attributes

In so far we assumed that different attributes affected co-occurrence in an independent fashion. Here we show that our main result holds, even when it is not true. We consider a more generic co-occurrence model with $P(i, j) = Z(i)Z(j) \prod_{k,k'} P^{(k,k')}(i, j)$ where $P^{(k,k')}(i, j) = 1 + s_{k,k'} \alpha_i^{(k)} \alpha_j^{(k')}$; and the $Z(i)$ are normalization factors chosen such that $\sum_j P(i, j) = P(i)$. To maintain the symmetry of the co-occurrence matrix requires $s_{k,k'} = s_{k',k}$.

Consider, to simplify notations, the symmetric $p_k = 1/2$ case. We have that $\sum_j \prod_{k,k'} P^{(k,k')}(i, j) = 1 + \mathcal{O}(s_{k,k'}^2)$; so at this order of approximation henceforth considered, we have: $P(i, j) \approx P(i)P(j) \prod_{k,k'} P^{(k,k')}(i, j)$ where $P(i) = 2^{-d}$. As a result, the PMI matrix for a pair of words, $(i, j)$, is

$$\text{PMI}(i, j) = \sum_{k,k'} \log(P^{(k,k')}(i, j)) \tag{31}$$

Each of the constituent $\log\left(P^{(k,k')}\right)$ can be re-expressed as

$$\log\left(P^{(k,k')}(a, b)\right) = \log(1 + s_{k,k'}ab) = \delta_{k,k'} + \gamma_{k,k'}ab \tag{32}$$

where $\delta_{k,k'} = \frac{1}{2}\left(\log(1 + s_{k,k'}) + \log(1 - s_{k,k'})\right)$ and $\gamma_{k,k'} = \frac{1}{2}\left(\log(1 + s_{k,k'}) - \log(1 - s_{k,k'})\right)$. Identifying $\alpha_i^{(k)}$ and $\alpha_j^{(k')}$ with $a$ and $b$, we have that

$$\text{PMI} = \boldsymbol{\delta} + \mathbf{A}\boldsymbol{\gamma}\mathbf{A}^\top \tag{33}$$

where $\boldsymbol{\delta} = \mathbf{1}\mathbf{1}^\top \sum_{k,k'} \delta_{k,k'}$ the rows of $A \in \mathbb{Z}_2^{2^d \times d}$ correspond to $\vec{\alpha}_i$ as before, and the elements of the $\boldsymbol{\gamma}$ matrix are just the $\gamma_{k,k'}$.

Thus the main conclusions of Section 6 hold: The rank of the PMI is at most $d+1$ and its eigenvectors are linear in the attributes, implying that linear analogies hold exactly. This is the central result. However, because $\boldsymbol{\gamma}$ is not diagonal, eigenvectors of the PMI will linearly combine attributes.

# NeurIPS Paper Checklist

The checklist is designed to encourage best practices for responsible machine learning research, addressing issues of reproducibility, transparency, research ethics, and societal impact. Do not remove the checklist: **The papers not including the checklist will be desk rejected.** The checklist should follow the references and follow the (optional) supplemental material. The checklist does NOT count towards the page limit.

Please read the checklist guidelines carefully for information on how to answer these questions. For each question in the checklist:

- You should answer [Yes] , [No] , or [NA] .
- [NA] means either that the question is Not Applicable for that particular paper or the relevant information is Not Available.
- Please provide a short (1–2 sentence) justification right after your answer (even for NA).

**The checklist answers are an integral part of your paper submission.** They are visible to the reviewers, area chairs, senior area chairs, and ethics reviewers. You will be asked to also include it (after eventual revisions) with the final version of your paper, and its final version will be published with the paper.

The reviewers of your paper will be asked to use the checklist as one of the factors in their evaluation. While "[Yes] " is generally preferable to "[No] ", it is perfectly acceptable to answer "[No] " provided a proper justification is given (e.g., "error bars are not reported because it would be too computationally expensive" or "we were unable to find the license for the dataset we used"). In general, answering "[No] " or "[NA] " is not grounds for rejection. While the questions are phrased in a binary way, we acknowledge that the true answer is often more nuanced, so please just use your best judgment and write a justification to elaborate. All supporting evidence can appear either in the main paper or the supplemental material, provided in appendix. If you answer [Yes] to a question, in the justification please point to the section(s) where related material for the question can be found.

IMPORTANT, please:

- **Delete this instruction block, but keep the section heading "NeurIPS Paper Checklist",**
- **Keep the checklist subsection headings, questions/answers and guidelines below.**
- **Do not modify the questions and only use the provided macros for your answers**.

