# OpenReview forum: "On the Emergence of Linear Analogies in Word Embeddings"
_NeurIPS.cc/2025/Conference — NeurIPS 2025 poster_

### Official Review · Reviewer_imGd · 2025-06-19

**Clarity:** 3
**Significance:** 3
**Originality:** 3
**Rating:** 5
**Confidence:** 3

**Summary:**

Summary: This paper proposes a latent factor generative model to explain the emergence of linear analogies in word embeddings. Specifically, it hypothesizes that each word is defined by a vector of latent binary attributes (eg: feminine v masculine, adj vs noun, etc), and that word co-occurrence probability depends on the shared attributes. It then follows this through to study how word embeddings and analogies would arise under such an assumption, and compares the predictions of this model to empirical observations.  The results show various ways in which this model makes better predictions than prior theoretical models for word embeddings. For example, in terms of the PMI spectrum, the embedding dimension required for analogies to arise, robustness to pruning specific co-occurrence examples from the training set, etc.

**Questions:**

- What can this tell us about popular contemporary LLMs?
- Does this analysis give us any insight into how we can develop better word embeddings or improve LLM design today?
- Is it possible to do some kind of binary latent factor analysis to discover the actual attributes that might underpin the existing words?

**Ethical Concerns:**

["NO or VERY MINOR ethics concerns only"]

**Final Justification:**

Thanks to the authors for the rebuttal comments. I have read all the reviews and author feedback. I don't see any dealbreakers for this work, it seems worth acceptance.

**Limitations:**

Limitations are OK.

**Paper Formatting Concerns:**

The references are a bit messy and should be made more consistent in formatting. Some of the author names seem to be wrong, eg [26,24].

**Quality:**

3

**Strengths And Weaknesses:**

Strengths:
+ This paper provides an elegant and intuitive generative model that could explain word embeddings and analogies.
+ This is more elegant than previous theoretical studies that attempted to explain analogies, but didn’t provide a complete generative model.
+ Predictions by the proposed model match empirical data quite well.

Weaknesses:
- Polysemous words, hierarchy, etc are not so neatly explained by this model (but this is a common limitation in prior work as well).
- The simple co-occurence based models mentioned (glove, w2v, etc) are rather old by today’s standards.

---

> ### Author Rebuttal · Authors · 2025-07-30
>
> Referee imGd
>
> We thank the referee for their comments and questions.
>
> Weaknesses
>
> 1. Polysemous words, hierarchy, etc are not so neatly explained by this model (but this is a common limitation in prior work as well).
>
> Polysemous words could be studied theoretically in our framework by ‘merging’ pairs of semantically distinct words (i,j) into a polysemous word k, by summing their corresponding rows and columns in the PMI matrix, such that P(ell)= P(i)+P(j) and P(w, ell)= P(w,i)+P(w,j) for any word w. It deserves a study of its own, but we expect our results - which are extremely robust to perturbations - to hold if the fraction of polysemous words is small.
>
> Modeling the presence of hierarchical attributes is a nice problem for the future; we hope our contribution will stimulate future studies in that direction.
>
>
> 2. The simple co-occurence based models mentioned (glove, w2v, etc) are rather old by today’s standards.
>
> See below.
>
>
> Questions
>
> 1. What can this tell us about popular contemporary LLMs?
>
> See reply to referee Zp5k to weaknesses 4+9: linear analogies are also observed in LLMs. They are argued to be functionally important, as they can mechanistically explain LLM fact retrieval in some cases (Merullo et al., ACL 2024).  How they emerge is not fully understood, and our work is a first step in seeking to build this understanding. We will discuss this literature in the introduction.
>
>
> 2. Does this analysis give us any insight into how we can develop better word embeddings or improve LLM design today?
>
> It was shown with SENSEBERT that if a machine (BERT in that case) is trained by predicting the meaning (a linguistic notion; e.g. cat belongs to the “animal” category) of a word and not only the word itself, performance is improved (Levine et al., ACL 2020).  It would be very interesting to check if LLMs perform better if instead of predicting the next token, they are trained to also predict the attributes of the associated word (see below for a practical proxy to these attributes).
>
>
> 3. Is it possible to do some kind of binary latent factor analysis to discover the actual attributes that might underpin the existing words?
>
> As a proxy for attributes, we could simply consider the projections of word embeddings into the top eigenvectors of the PMI matrix. As discussed in the text and above, attributes are somewhat mixed in these eigenvectors. However, linear combinations of attributes may be enough for practical purposes, such as the extension of SENSEBERT discussed just above. We will discuss these interesting points in the conclusion.

---

> > ### Comment · Reviewer_imGd · 2025-08-04
> > **Thanks**
> >
> > Thanks to the authors for the rebuttal comments.
> > I have read all the reviews and author feedback.
> > I don't see any dealbreakers for this work, it seems worth acceptance.

---

### Official Review · Reviewer_vqG4 · 2025-06-20

**Clarity:** 3
**Significance:** 2
**Originality:** 2
**Rating:** 5
**Confidence:** 3

**Summary:**

This paper presents a theoretical generative model aimed at explaining why linear analogies (such as $W_{\text{king}} - W_{\text{man}} + W_{\text{woman}} \approx W_{\text{queen}}$) naturally arise in word embeddings like Word2Vec and GloVe. Key contributions include:

1. **Generative Model:** Proposes that words can be represented through binary semantic attributes, with co-occurrence probabilities determined by independent interactions between these attributes. This creates a structured co-occurrence matrix enabling analytic computation of eigenvectors and eigenvalues.

2. **Emergence of Analogies:** Demonstrates analytically that linear analogies naturally emerge due to dominant eigenvectors of the co-occurrence (or PMI) matrix, directly reflecting arithmetic relationships between word attributes.

3. **Robustness Analysis:** Validates the theory numerically, showing that the linear analogy structure is robust against noise, sparsity (subsampling vocabulary), and even the removal of analogy-specific word pairs.

4. **Empirical Validation:** Tests predictions using actual co-occurrence data from Wikipedia, showing strong agreement between theoretical predictions and empirical observations.

Overall, this paper provides a rigorous theoretical explanation for the presence and robustness of linear analogy structures observed empirically in widely used word embeddings.

**Questions:**

N/A

**Ethical Concerns:**

["NO or VERY MINOR ethics concerns only"]

**Final Justification:**

The authors clarified my concerns.

**Limitations:**

yes

**Quality:**

2

**Strengths And Weaknesses:**

### Strengths:
1. Theoretical Clarity and Rigor
2. Explanatory Power
3. Robustness Analysis
4. Empirical Validation

### Weaknesses:
1. Simplified Assumptions: The model makes strong simplifying assumptions, including binary semantic attributes and independence of attribute effects.
2. Generalizability and Realism: The assumption of all possible combinations of attributes existing in the vocabulary is unrealistic.
3. Potential Degeneracies in Parameter Choices: The model’s predictive accuracy and analogy results are sensitive to the choice and distribution of parameters (e.g., semantic strength distributions $s_k$).

---

> ### Author Rebuttal · Authors · 2025-07-30
>
> Referee vqG4
>
> We thank the referee for their useful comments.
>
> Weaknesses
>
> 1. Simplified Assumptions: The model makes strong simplifying assumptions, including binary semantic attributes and independence of attribute effects.
>
> See response to referee Zp5k:  we had already provided an extension to tertiary semantic attributes in the appendix. Extension to n-valued attributes is straightforward. In response to both referees, we have extended our theory to the case of correlated attributes, see a sketch of the formalism above, which will appear in Appendix in the final paper.
>
>
> 2. Generalizability and Realism: The assumption of all possible combinations of attributes existing in the vocabulary is unrealistic.
>
> (i) Our model is robust to sparsification  (i.e. subsampling the vocabulary). In that case, all possible combinations of attributes do not exist in the vocabulary. Thus, we have already tested that this assumption can be relaxed.
> (ii) Clearly, some attributes do not enter in the semantic meaning of a given word. For example, a word can be neither masculine nor feminine. This can be readily modeled in our framework by considering tertiary attributes with a “neutral” category, each time a given attribute does not apply to a given word.
> We will add a discussion on these points.
>
> 3. Potential Degeneracies in Parameter Choices: The model’s predictive accuracy and analogy results are sensitive to the choice and distribution of parameters (e.g., semantic strength distributions ).
>
> The notion that there exists a distribution of semantic strength is not a bug in our opinion, but a nice feature of our approach. It explains why distinct linear analogies do not appear at the same dimension of embedding, which is not captured by previous models.

---

> > ### Comment · Reviewer_vqG4 · 2025-08-04
> >
> > Thank you for your clarifications! I have no further questions/comments and update my score accordingly.

---

### Official Review · Reviewer_Zp5k · 2025-07-05

**Clarity:** 2
**Significance:** 3
**Originality:** 3
**Rating:** 3
**Confidence:** 2

**Summary:**

This paper proposes a theoretical generative model for word co-occurrences. By defining each word as a vector of d binary semantic attributes and assuming that co-occurrence probabilities result from independent attribute interactions, the paper analytically derives the PMI and M matrices and systematically explains the mechanism behind the emergence of linear analogy relationships in word embeddings.
Beyond its core theoretical contribution, the paper also validates its predictions via extensive numerical simulations and confirms strong alignment with empirical statistics from real corpora like Wikipedia. Notably, the model is shown to be robust to noise, vocabulary sparsification, and even the removal of specific analogy word pairs.

**Questions:**

1. Could correlated semantic attributes bias the model’s predictions or its spectral structure?
2. Has the model been evaluated beyond English corpora (e.g., on morphologically rich or logographic languages such as Chinese or Finnish)?
3. In the era of contextualized word representations (e.g., BERT, GPT), what is the broader significance of studying static word embeddings?
How sensitive is the model to the choice of the PMI transformation (e.g., using shifted PMI, SVD truncation, or subsampling strategies)?
4. Could you provide insights or explanations regarding the five weaknesses previously identified?
5. The model assumes attribute independence. Could you provide examples illustrating that such independence is satisfied within the Wikipedia dataset?
6. The physical meaning of some notations in Section 4 is confusing. Why does $p_k \leq 1/2$? And what is the significance of The symmetric matrices $P^{(k)}$, are they denoting probabilities, since the text states $\sum_j P(i, j) = P(j)$? However, the values in Equation (4) do not all lie within the [0, 1] interval—for instance, $1 + s_k > 0$—which is perplexing. Could the meaning of the formulas and symbols in Section 4 be clarified in greater detail?

**Ethical Concerns:**

["NO or VERY MINOR ethics concerns only"]

**Final Justification:**

I appreciate the valuable efforts of the authors in the response, which have addressed most of my concerns. Considering the overall quality of this work, I would like to keep the BR rating.

**Limitations:**

Yes.

**Paper Formatting Concerns:**

No concerns.

**Quality:**

2

**Strengths And Weaknesses:**

Strengths
1. This paper proposes a theoretical generative model for words based on binary semantic properties, directly linking the structure of co-occurrence or PMI matrices to the emergence of linear analogies, thus effectively explaining phenomena observed in prior work.
2. This paper not only provides a conceptual explanation, but also derives the spectral structure of the co-occurrence matrix through closed-form resolution. Moreover, by comparing the spectral properties of the model-generated matrix with empirical data from Wikipedia, the theoretical predictions are validated.
3. Code and data are provided to ensure reproducibility of the paper.

Weaknesses
1. The mathematical notation and statements used in the paper lack clarity. Some sections, particularly section 4, 5 and 6, use a lot of mathematical notation, the logical transitions between mathematical statements and their explanations are abrupt, and even some notations are not explained, e.g. $\delta_{a^{(k)}_i, 1}$. These paragraphs may be difficult to understand for researchers unfamiliar with these theories. The use of simpler and clearer language with appropriate examples may give the reader a better understanding.
2. The modeling assumptions have limitations. The theoretical model assumes that semantic properties are strictly binary and mutually independent. However, the continuous nature and context-dependent relevance of semantics in the real world mean this assumption diverges significantly from reality, which significantly limits the model's broader applicability.
3. This paper empirically validates on Wikipedia data and standard analogy benchmarks, demonstrating validity only on single corpus and synthetic data.  However, it was not validated on more complex corpora, such as social media text or multilingual data, and the results may not generalize to morphologically rich languages.
4. The conclusions focus on the classical embedding model (Word2Vec/GloVe), but do not discuss the connection with the Large Language Model (LLM). Do the latter produce linear analogy due to the same mechanisms?
5. The limited discussion of existing methods. Section III identifies the reasons why existing methods are problematic and whether more detailed elaboration or empirical results can be provided to quantify these problems.
6. Binary attributes may oversimplify continuous/multi-valued semantics.
7. The author assumes that  each attribute of a word affects its context in an independent manner. But real-world attributes may correlate, potentially underestimating complex interactions.
8. The paper lacks empirical comparison with alternative models to contextualize the novelty and practical benefit of the proposed framework.
9. Despite its theoretical depth, the model may have limited applicability to modern LLM embeddings.

---

> ### Author Rebuttal · Authors · 2025-07-30
>
> Referee Zp5k
>
> We thank the referee for their comments and criticism, they are useful to clarify the manuscript.
>
> Weaknesses:
>
> We respond to the distinct points raised by the referee below:
>
>
> 1. Summary of referee comment: The mathematical notation and statements used in the paper lack clarity, particularly in sections 4, 5, and 6.
>
> We will improve the flow and clarity, particularly in the highlighted sections. The $\delta$ refers to the standard Kronecker delta, so $\delta_{a_i^{(k)},1}$ evaluates to 1 whenever $a_i^{(k)}$ is 1, and is otherwise zero. We note that the appendix also offers a more pedagogical treatment of several of the calculations, a review of Kroenecker products, and an explicit worked example for $d=2$. We will draw the reader’s attention to this earlier in the manuscript.
>
>
> 2+6+7. Summary of referee comment: The model makes simplifying assumptions: (i) binary variables don’t capture multi-valued (or continuous) semantics, (ii) there may be correlations between semantic attributes.
>
> Points 2+6 criticize the assumption of binary variables. Please note that our model can be generalized to multi-valued semantics. In fact, we had already performed an extension to tertiary variables in the appendix, and our findings carry through largely unchanged. We will highlight this result earlier in the paper. The extension to n-valued variables is straightforward, and may represent an approach to the continuous limit. We hope our contribution inspires such lines of inquiry.
>
> Point 2 also points to the fact that semantic variables might be correlated.  This is an excellent comment, which led us to generalize our model to correlated attributes. We  briefly sketch this generalization below in answering Q1.
>
>
> 3. Validation on a single test set (Wikipedia), and limitations of applicability more broadly to other languages or corpora.
>
> The presence of linear analogies our theory seeks to explain has been demonstrated in numerous other languages, including more morphologically complex ones such as Finnish (Venekoski and Vankka, ACL 2017) and Chinese (Su and Lee, ACL 2017) when using the appropriate scale of morphologically aware tokenization, such as FastText (Bojanowski et al., ACL 2017). We will make sure to cite these references in our introduction, to highlight the generality of the problem we are addressing.
>
> The objective of our contribution is to generate a robust, minimal theoretical framework that explains several key aspects of word embeddings. Other work in the realm of theory building (e.g. Arora et al. TACL 2016) also uses a single dataset and language (English Wikipedia). Comparing quantitatively multiple languages is certainly an interesting endeavour, but it presents a work on its own. It should not obscure the presentation of the new formalism introduced here in its purest form.
>
>
>
> 4+9. Summary of referee comments: This work addresses primarily a feature of context-free word-embedding models. To what extent does this generalize to LLMs?
>
> The semantic content of language exhibits both elements that are independent of context, and others that depend on it (HP Grice 1975, Pustejovsky, J., 1995). Strictly speaking, our theory is making statements about the former, since it studies context-free embeddings like word2vec. Concerning LLMs, they must necessarily also capture this context-free element in their internal representations, and so our work has some bearing on them. Bommasani et al. 2020 showed that such context-free embeddings can be obtained by pooling the representation of words across many contexts. This approach could be used to test our views in LLMs, as we will discuss in the conclusion.
>
> The work of Hernandez et al. 2024 ICML is also very pertinent: The authors find that the internal representations of LLMs obey linear analogy (up to a context-dependent linear projection), satisfying relations like $W_r (\text{Miles Davis} - \text{Billy Joel}) = \text{Trumpet} - \text{Piano}$, where the relational matrix $W_r$ is induced by the words “plays the instrument” (and thus depend on context). Thus, linear analogy structure survives in the inner representations of LLMs in a contextual way. Other work by Merullo et al., ACL 2024, indicate that these linear analogies are important for fact retrieval by LLMs. Understanding the emergence of contextual linear analogies remains a theoretical challenge; the ideas presented in our work are possible handles to tackle it.  We will discuss these works in conclusion.
>
>
>
> 5+8. Limited comparison between our work and existing methods.
>
> The limited comparison with existing methods that the reviewer points out reflects the novelty of our work: we are making predictions where other approaches did not, including on how the quality of linear analogies depends on the dimension of the embedding considered. To our knowledge, there are no alternative predictions in the literature on that. We simply emphasize in the text that some key assumptions of previous methods do not hold (such as the isotropy of the PMI matrix).
>
>
>
> Questions:
>
> 1. Could correlated semantic attributes bias the model’s predictions or its spectral structure?
>
> We thank the referee for this interesting point. It has prompted us to consider an extension to the model that includes correlated attributes, which we will include in the appendix of the paper. The PMI matrix is robust to correlations.
>
> In brief, in this extension equation 3 takes the modified form:
> $$ P(i,j) = P(i)P(j) \left( \prod_k P^{(k)}(i,j) \right) \left( \prod_{k, k’} P^{(k, k’)}(i,j) \right) $$
> where the $P^{(k,k’)}(i,j) = 1 + s_{k,k’} \alpha_i^{(k)}\alpha_j^{(k’)}$ introduces correlation between attributes $k$ and $k’$ of strength $s_{k,k’}$. This formulation approximately preserves normalization when the $s_{k,k’}$. The subsequent analysis of the PMI matrix carries through in the same manner, resulting in an identical factorization with equation 12, except that the matrix $D$ is no longer diagonal. All of the following results (1,2,3,4) hold, along with the emergence of linear analogies at dimension $d$. The main difference is that the eigenvectors of the PMI matrix in this case do not map one-to-one onto semantic attributes, but this has no impact on the linear analogies.
>
>
> 2. Has the model been evaluated beyond English corpora (e.g., on morphologically rich or logographic languages such as Chinese or Finnish)?
>
> See our response to weakness 3 above.
>
>
> 3. In the era of contextualized word representations (e.g., BERT, GPT), what is the broader significance of studying static word embeddings?
>
> See our response to weakness 4+9 above.
>
>
> How sensitive is the model to the choice of the PMI transformation (e.g., using shifted PMI, SVD truncation, or subsampling strategies)?
>
> (i) Shifted PMI adds a constant offset to the PMI matrix. This is a rank-1 perturbation to the PMI matrix and  therefore has minimal effects on its spectral properties, as we checked.
> (ii) The choice of the embedding space dimension $K$ already implements SVD truncation.
> (iii) Regarding subsampling: we test a subsampled (sparsified) vocabulary in figure 2, panel f in the model. We have checked that subsampling by limiting the vocabulary also does not affect our results on Wikipedia, as we will mention in the text.
>
>
> 4. Could you provide insights or explanations regarding the five weaknesses previously identified?
>
> See above.
>
>
> 5. The model assumes attribute independence. Could you provide examples illustrating that such independence is satisfied within the Wikipedia dataset?
>
> As we highlight in our response to question 2 and will add to the paper, our theoretical model is robust to including a correlation between attributes.

---

### Decision · Program_Chairs · 2025-09-17

**Decision:**

Accept (poster)

**Comment:**

The paper proposes a generative model to explain linear analogies in word embeddings.  This is done by hypothesizing that each word is a vector of binary attributes and that word co-occurrence probability depends on the shared attributes.  Under this assumption the authors study the emergence of embeddings and analogies and their predictions are compared with practice.  The results of the authors show various ways in which the proposed model makes better predictions than prior theoretical models for word embeddings.

It is unfortunate but only three out of the four reviewers returned a review.  Furthermore, the reviewer who is recommending borderline reject, probably busy with other things, was also slow in responding to the authors' rebuttal and explain what other parts required elaboration from the point of view of the authors.  This way the authors never had a chance to make an additional comment to the reviewer's comment on the authors' rebuttal.  Admittedly, most of the concerns of Reviewer Zp5k were addressed by the authors, but the main issues that remained for the reviewer were:

(1) the empirical validation was limited to English Wikipedia, and

(2) the connection to modern contextualized models is largely theoretical and untested.

I think that Issue (1) about the use of the English language is minor.  In my opinion the authors give a good answer to defend this issue in Point (3) of their rebuttal.  Regarding Issue (2), I believe there can be two "camps" given the wild enthusiasm that LLMs enjoy in recent years.  In my opinion the authors result have merit even if they do not directly touch LLMs and moreover, the authors are willing to better frame their work in the final manuscript and clarify differences and similarities along this path.

Hence, at this point I have looked into the discussions, questions, arguments, and responses, and I have also looked into the paper.  I am more in favor of suggesting acceptance of this paper, rather than rejection.  So, this will be my recommendation, albeit I must stress that I am not entirely confident for my recommendation.